# Uptake and Advanced Therapy of Butyrate in Inflammatory Bowel Disease

**Shinji Ota and Hirotake Sakuraba ***

Department of Gastroenterology and Hematology, Hirosaki University Graduate School of Medicine, Hirosaki 036-8216, Japan
* Correspondence: hirotake@hirosaki-u.ac.jp; Tel.: +81-172-39-5053

**Abstract:** The pathogenesis and refractory nature of inflammatory bowel disease (IBD) are related to multiple factors, including genetic factors, environmental factors, and abnormalities in gut microbial diversity, which lead to decreased levels of short-chain fatty acids (SCFAs). Among SCFAs, butyrate plays an important role in mucosal barrier maintenance, serves as an energy source in intestinal epithelial cells (IECs), and exhibits anti-inflammatory effects; therefore, it is a particularly important factor in gut homeostasis. Changes in gut microbiota and butyrate levels affect the outcomes of drug therapy for IBD. Butyrate is mainly absorbed in the large intestine and is transported by monocarboxylate transporter 1 (MCT1) and sodium-coupled monocarboxylate transporter 1 (SMCT1). During gut inflammation, butyrate utilization and uptake are impaired in IECs. Dysbiosis and low abundance of butyrate affect fecal microbiota transplantation and anticancer immunotherapy. Although butyrate administration has been reported as a treatment for IBD, its effects remain controversial. In this review, we discuss butyrate absorption and metabolism in patients with IBD and their relationship with drug therapy.

**Keywords:** inflammatory bowel disease; butyrate; microbiota; intestinal epithelial cells; monocarboxylate transporter 1

## 1. Introduction

Ulcerative colitis (UC) and Crohn's disease (CD) are considered the major forms of inflammatory bowel disease (IBD) and chronic inflammation in the intestine; however, their etiology is not fully understood. IBD onset is believed to be related to genetic factors, diet, and dysbiosis. IBD has the highest occurrence in developed countries, such as North America and Europe, and has recently been increasing in Asia [1,2]. Japan, Hong Kong, and Korea have an increased incidence of IBD. Regardless of genetic background, the number of patients with IBD has increased. This phenomenon suggests that the rapid increase in the incidence of IBD is parallel to changes in lifestyle, such as diet, breastfeeding, and antibiotic use [3]. Additionally, these parameters are closely related to the gut microbiota. Patients with IBD demonstrate significantly decreased microbial diversity and short-chain fatty acids (SCFAs) compared with healthy controls [4–6]. This suggests that environmental factors, such as microbiota and diet, contribute to abnormal immune responses during the onset of IBD.

Recent studies have shown that microbial diversity and fecal abundance of butyrate influence the success of biological therapies and thiopurine treatment [7–10]. These results suggest that interactions between intestinal dysbiosis and the host response are involved in both the onset of IBD and treatment outcomes. In this review, we describe the multiple effects and advanced therapies of butyrate, as well as the regulation of butyrate uptake into intestinal epithelial cells (IECs).

## 2. SCFAs and Intestinal Homeostasis in IBD

### 2.1. The Role of SCFAs in Intestinal Immune Response

SCFAs are metabolites produced by the intestinal microbiota and are mainly composed of acetate, propionate, and butyrate. SCFAs have various physiological effects in humans and play important roles in colon homeostasis [11]. SCFAs have many favorable effects, including their barrier function, serving as an energy source of IEC, activating G-protein coupled receptors (GPCRs), and inhibiting the effects of histone deacetylase (HDAC) on cell differentiation and proliferation [12]. GPCRs include GPR41, GPR43, and GPR109A; anti-inflammatory effects in the intestinal mucosa are due to HDAC inhibition and GPCR activation [13]. GPR41 is preferentially expressed in the adipose tissue, while GPR43 and GPR109A are expressed in IECs and immune cells. SCFAs modulate the immune response [13]. In particular, butyrate strongly induces regulatory T cell (Treg) differentiation by enhancing the production of TGF-β in the colon via acetylation of the Foxp3 promoter in naive T cells [14,15]. Butyrate also promotes IL-22 production from CD4+ T cells and innate lymphoid cells (ILCs); IL-22 plays an important role in maintaining the health of the epithelial barrier [16]. Moreover, butyrate, but not other SCFAs, enhances mucosal barrier function through actin-associated gene expression and AMP-activated protein kinase [17,18]. In colonic cell lines, by activating GPR109A, butyrate suppressed inflammation due to LPS-induced NF-kβ activation [19]. Butyrate suppressed the production of proinflammatory cytokines from neutrophils of patients with IBD and ameliorated inflammation in an animal colitis model by regulating neutrophils [20]. Thus, butyrate plays a major role in suppressing gut inflammation and maintaining epithelial barrier function.

### 2.2. SCFAs in IBD

Dysbiosis is generally accepted to be involved in IBD onset. Diet plays a role in gut homeostasis and influences gut microbiota composition, intestinal permeability, and mucosal barrier [21]. Westernized diets contain high sugar, high fat, and low fiber levels and high calories; they elevate the risk of IBD onset [22,23]. Butyrate is produced mainly by bacteria in the human colon, particularly *Faecalibacterium prausnitzii*, *Eubacterium rectale*, and *Roseburia hominis* [17,18]. Patients with IBD have diminished microbial diversity, that is, the abundance of *F. prausnitzii* and *R. hominis* is reduced [24,25]. Furthermore, IECs metabolism plays an important role in regulating the gut microbiota. In a healthy state, IECs utilize butyrate as an important energy source. Since butyrate metabolism consumes oxygen, surface colonocytes are hypoxic, which promotes the growth of anaerobic bacteria such as *Firmicutes* in the lumen. Gut inflammation leads to the reduction of *Firmicutes* and butyrate levels. In the absence of butyrate, IECs obtain energy through fermentation metabolism of glucose; consequently, oxygen concentration in the lumen increases to 3–10%, and it alters the local microbial composition [26]. Its decreases strict anaerobes while allowing the growth of facultative anaerobes such as *Escherichia coli* pathobionts, which are believed to be involved in IBD [26,27].

## 3. Butyrate Transport to IECs

Butyrate is a weak acid (pKa = 4.8), and more than 90% exists in an ionized form in the colon lumen; therefore, carrier-mediated transport to IECs is a fundamental pathway [28–31]. Butyrate exists mainly in the proximal colon [31]. Monocarboxylate transporter 1 (MCT1), encoded by SLC16A1, and sodium-coupled monocarboxylate transporter 1 (SMCT1), encoded by SLC5A8, are expressed in the apical membrane of colonocytes; these are major transporters involved in butyrate absorption [11]. Butyrate uptake mediated by MCT1 is based on Michaelis–Menten kinetics in the 4–10 mM range [32,33]. In contrast, SMCT1 is based on Michaelis–Menten kinetics in the low micromolar range (50–100 μM) [11]. The concentration range of butyrate in the human colon is 10–20 mM [31]. Therefore, SMCT1 is mostly saturated under normal conditions. Butyrate is absorbed by IECs and is metabolized as an energy source. Butyrate undergoes β-oxidation in mitochondria and enters the tricarboxylic acid (TCA) cycle, leading to energy production (Figure 1) [34]. This mechanism provides 70–80%

of the IEC energy requirements [28]. Decreased butyrate absorption in IECs causes autophagy via AMPK activation [35]. In contrast, unmetabolized butyrate inhibits histone deacetylase (HDAC) activity and is partially transported to the basolateral side by MCT4 [11,36]. Systemic butyrate concentration in human peripheral blood is 1–10 μM [31,37,38].

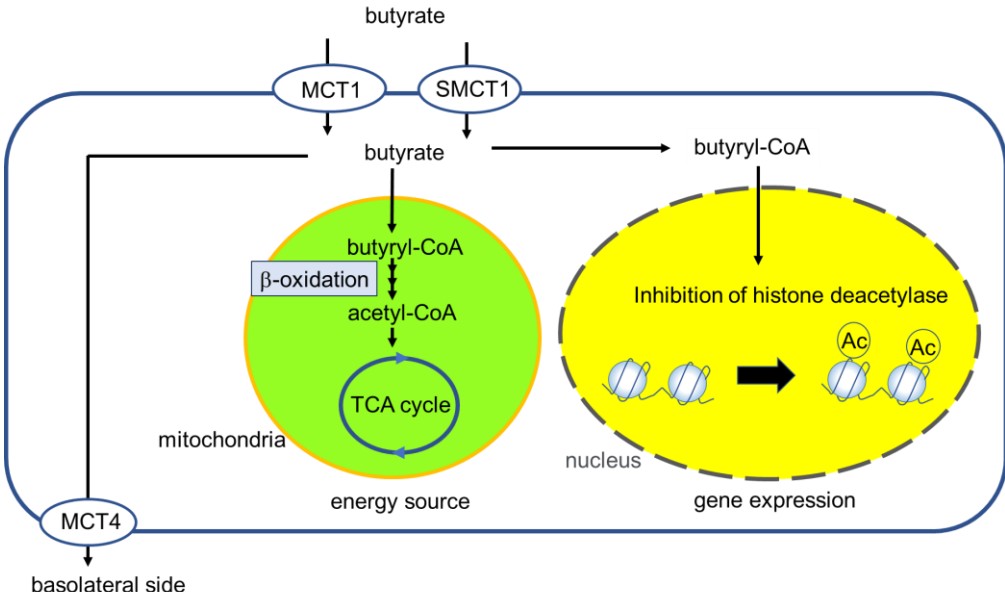

**Figure 1.** Overview of butyrate uptake, dynamics, and physiological effects in intestinal epithelial cells (IECs). Butyrate fermented by microbiota is transported to IEC by monocarboxylate transporter 1 (MCT1) and sodium-coupled monocarboxylate transporter 1 (SMCT1). By β-oxidization in the mito-chondria, butyrate is metabolized to acetyl-CoA. Unmetabolized butyrate inhibits histone deacetylase and is transported into the lamina propria.

## 4. Regulation and Role of MCT1 in IECs

Luminal butyrate concentration can be modulated by diet, gut inflammation, and dysbiosis. Butyrate itself enhances MCT1 expression via the NF-kβ pathway and mRNA stabilization [39,40]. Abundance of fiber in the diet influences its absorption. In rats and pigs fed with pectin or resistant-starch diets, the expression of MCT1 was increased in the colon [41,42]. Moreover, luminal butyrate activates GPR109A, which is also reported as an SCFA sensor and expressed in the apical membrane of IECs. Activated GPR109A enhances membrane MCT1 expression via decreased cAMP pathway [43]. As a result, Vmax of MCT1-mediated butyrate uptake is increased, which leads to efficient absorption [43]. Previous studies have reported the membrane expression of MCT1 without changes at the transcriptional level. Although precise mechanisms are not known, including the recycling mechanism and protein trafficking systems, calcineurin has been suggested to regulate cAMP degradation and affect cytoskeletal function [33,44,45]. In IBD patients, intestinal MCT1 protein expression is downregulated [46]. This is considered to be due to a decrease in butyrate-producing bacteria and inflammation itself. Interferon γ (IFN-γ) and TNF-α signaling downregulate the expression of MCT1 in HT-29 cells [46]. In pig colonic tissue and Caco-2 cells, TNF-α downregulated MCT1 expression [47]. Thus, MCT1 expression and function are dynamically altered by intestinal butyrate and inflammation. Moreover, in the intestinal mucosa of active UC, the intracellular butyrate availability is reduced due to the downregulation of β-oxidation pathway [48]. Hence, inflammation impair the butyrate uptake and metabolism in IECs, followed by the disruption of gut homeostasis.

In a previous study, the ability of organoid epithelial cells to perform butyrate uptake, oxidization, and metabolism in patients with non-active IBD was comparable to that in healthy individuals; however, in patients with active IBD and upon TNF-α stimulation, this ability was decreased [13,49]. Thus, in the active inflammatory state in IBD, the uptake and

utilization of butyrate are decreased, even if abundant butyrate is available from microbiota or supplementation.

## 5. Gut Microbiota and Systemic Immune Response

The development of new therapeutic agents based on an advanced understanding of IBD pathophysiology has resulted in favorable outcomes and reduced disease-related morbidity. The recently proposed treat-to-target (T2T) strategy has shown that early initiation of advanced therapies, such as anti-TNF-α agents, leads to greater response. As per this concept, personalized treatment according to clinical poor prognostic factors is key. However, prediction of the treatment response to each therapeutic drug based on the mode of action is not yet established. In recent years, the association between the intestinal microbiota and various immune-mediated diseases has attracted attention. Several studies have highlighted the role of gut microbiota and metabolites in the response to treatments, including biologics, fecal microbiota transplantation (FMT), and anti-cancer immunotherapy. Restoration of intestinal dysbiosis reduced diversity, and butyrate abundance by FMT directly targets dysbiosis. Successful treatment with FMT has been associated with the presence of *Eubacterium* and *Roseburia* species, SCFA biosynthesis, and secondary bile acids. In contrast, failure of FMT treatment has been associated with *Fusobacterium*, *Sutterella*, and *Escherichia* species [50]. Previous studies have shown a correlation between gut microbiota and anti-cancer immunotherapy, such as cyclophosphamide [51]. Specific species of bacteria induce Th1 cell immune responses against lung and ovarian cancers and predict long-term anticancer responses. The gut microbiota is believed to affect the tumor microenvironment (TME) to regulate anticancer effector T cells and cytokine responses.

In patients with IBD, gut microbes and high butyrate levels predict a better response to biologics and azathioprine (Table 1) [7–10]. However, the precise pathway by which the gut microbiota affects the response to biological therapies remains obscure. One mechanism hypothesized is that a low level of decreased diversity and butyrate abundance minimizes mucosal barrier disruption, because of which immunomodulatory therapy is expected to easily succeed. Further studies are required to uncover the full mechanistic pathway by which the gut microbiota and its metabolites improve the response to advanced therapies for IBD.

**Table 1.** Effects of butyrate and butyrate-producing bacteria on therapeutic efficacy of drugs.

| Drug | Patients | Outcome | Ref. |
|---|---|---|---|
| anti-TNF-α | 56 UC (50 IFX, 6 ADA) | Lower dysbiosis indexes and higher abundance of *Faecalibacterium prausnitzii* predict responders | [10] |
| vedolizumab | 43 UC, 42 CD | Butyrate producer promote response of vedolizmab treatment | [7] |
| anti-TNF-α | 18 CD, 17 UC | In low response patients, lower exchanged among bacterial communities of butyrate, ethanol or acetaldehyde (involved in butyrate synthesis) | [9] |
| azathioprine | 43 CD, 22 UC | Butyrate production of patients in remission is at baseline higher compared with patients without remission | [8] |

## 6. Influence of Gut Microbiota on Response to Advanced Therapies

Maier et al. reported that 24% of 1197 drugs against 40 representative gut microbiota strains affect the growth of bacteria, thus modifying the efficacy and side effects of the drugs [52,53]. Butyrate ameliorates gut inflammation through multiple pathways and regulates the mucosal barrier function. Butyrate is considered the principal ATP source for IECs [54]. IEC obtained by biopsy from patients with UC shows impaired butyrate β-oxidation [55,56]. For example, it results from decreased expression of mitochondrial

acetoacetyl-CoA thiolase activity [57]. Butyrate enhances mucosal barrier function; however, the barrier is disrupted during active inflammation [58]. The expression of MCT1 in the colon is reportedly decreased in patients with IBD and experimental colitis models [46,47,59]. In patients with active IBD, butyrate utilization and uptake are impaired (Figure 2). The low amount of butyrate and other abnormalities of the microbiota are speculated to contribute to the refractory disease, even if drugs suppress inflammatory cytokines and immune cells. We reported that the efficacy of cyclosporine treatment in an animal colitis model deteriorated with antibiotic pretreatment but recovered with butyrate supplementation [60]. We showed that cyclosporine treatment upregulates the expression of membrane MCT1 in IECs without inducing transcriptional changes. Regarding the efficacy of drugs for IBD, focusing on butyrate absorption and abundance may be necessary. Butyrate exhibits a direct antimicrobial effect, for example, on *Acinetobacter baumannii*, *E. coli*, and *Staphylococcus pseudintermedius*, by inducing bacterial membrane depolarization and cytosolic acidification [61]. Moreover, a previous report showed that butyrate induced antimicrobial peptides, such as LL-37, in IECs in vivo and colon cell lines [62]. Thus, butyrate improves gut microbiota.

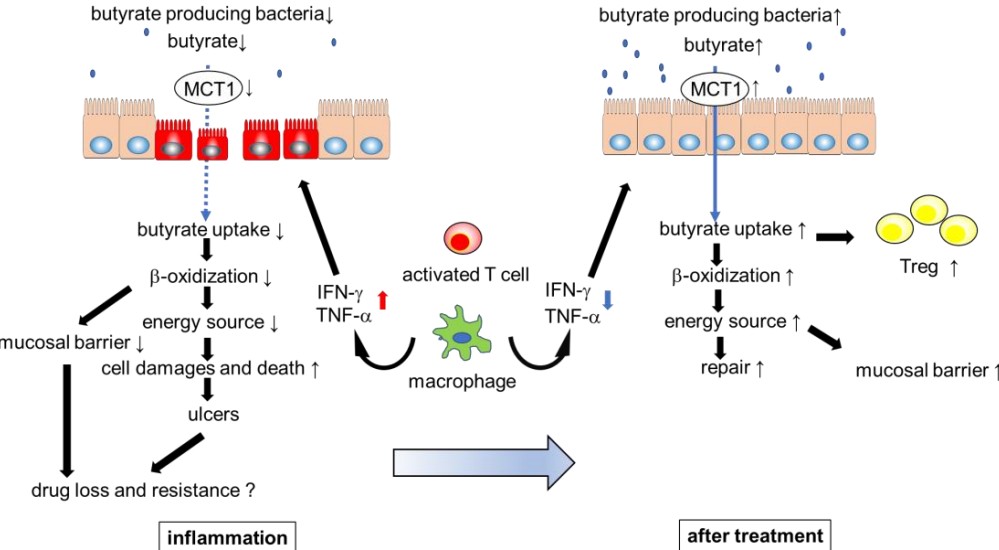

**Figure 2.** Inflammatory cytokines impair butyrate uptake, barrier function, and metabolism in intestinal epithelial cells (IECs). Inflammation and low abundance of butyrate contribute to the refractory nature of inflammatory bowel disease. Appropriate treatment ameliorates the inflammation that leads to dysbiosis and decreased butyrate uptake. As a result, increased butyrate levels in the colon contribute to gut homeostasis and drug efficacy.

Additionally, the gut microbiota is influenced by drug therapy. 5-Aminosalicylic acid (5-ASA) treatment altered the balance between increase in *Proteobacteria* and decrease in *Firmicutes* in inflamed mucosa in patients with UC [63]. Furthermore, after anti-TNF-αtherapy for patients with CD, the phyla *Proteobacteria* and *Clostridia* increased, and α-diversity was increased in anti-TNF-α therapy responders [64]. The *F. prausnitzii/E. coli* ratio was associated with the responder and non-responder status; it served as a biomarker of response to anti-TNF-α therapy [64].

## 7. Advanced Therapy with Butyrate

Treatments focusing on the supply of butyrate and intestinal microflora to the colon have been reported. Several reports have described that butyrate administration improves gut inflammation and relieves symptoms and microbiota changes in patients with IBD [65,66]. However, oral intake of butyrate is unacceptable because of its offensive odor and absorption in the small intestine. There have been several reports on butyrate modification. Tributyrin overcomes this disadvantage [67]. It is an ester composed of butyric acid and glycerol and is used as a food

additive. However, till data, no clinical reports indicate that tributyrin improves the condition of patients with IBD [68]. Butyration of starch has been showed to enable effective delivery to the colon [69,70]. Butyrylated starch is resistant to amylolysis in the small intestine and is released in the colon by bacterial esterase [71]. It has shown good efficacy in multiple mouse models of DSS-induced colitis [72]. Mu et al. reported polyvinyl butyrate nanoparticles that were efficiently delivered to intestinal macrophages and were effective against mouse DSS colitis [73]. Clinical trials in patients with IBD are expected.

In patients with mild to moderate UC, oral therapeutics of *Firmicutes* spores, which were purified from the stool of healthy donors by ethanol treatment, achieved a higher clinical remission rate compared with the placebo [74]. Butyrate did not change significantly after treatment; however, the ratio of butyrate to propionate tended to increase.

Oral treatment with microencapsulated butyrate increased the abundance of SCFA-producing microbiota, and with add-on therapy, maintained remission in patients with IBD [66,75].

However, some reports showed that butyrate treatment not only lacks efficacy in patients with IBD but also worsens colitis (Tables 2 and 3). Organoid culture sampling from the intestines of patients with UC showed that butyrate worsens barrier function and inflammatory cytokine expression under co-stimulation with TNF-$\alpha$ and IFN-$\gamma$ [76]. Kaiko et al. reported that butyrate prevents wound repair of intestinal stem cells [77]. Butyrate is potentially toxic at very high concentrations in the colon lumen [78]. Furthermore, reduction in luminal pH is associated with inhibition of the metabolic activity of luminal bacteria, and it can potentially have detrimental effects on other functions of the gut microbiota [68,79]. The luminal concentration of SCFA and the pH profile achieved by these compounds is unknown [68]. Butyrate treatment for IBD may be controversial because of the severity of inflammation in patients participating in clinical studies. Studies have suggested that because of the reduction in butyrate transporter levels, treatment with butyrate fails to improve mucosal damage in patients with active IBD.

**Table 2.** Clinical trial results for topical butyrate treatment.

| Disease | Sample Size | Treatment | Dose, Duration | Outcome | Ref. |
|---|---|---|---|---|---|
| distal UC | 12 | SCFAs (80:30:40) | 100 mL, twice daily, 6 weeks | 9 patients improved disease activity index and mucosal histrogical scores | [80] |
| | 10 | 100 mM butyrate vs. placebo | 100 mL, twice daily, 2 weeks | Improving endoscopic and histrogical scores | [81] |
| | 47 | 130 mM SCFAs (46:23:31) or 100 mM butyrate alone vs. saline | 60 mL, twice daily, 8 weeks | No differences in disease activity, clinical response, endoscopic, histological scores | [82] |
| | 40 | 150 mM SCFAs (53:20:27) vs. saline | 100 mL, twice daily, 6 weeks | Improving intestinal bleeding, urgency, the patient self-evaluation scores | [83] |
| | 38 | 80 mM butyrate vs. saline | 60 mL, once nightly, 6 weeks | No difference in clinical disease activity index scores | [84] |
| | 103 | 150 mM SCFAs (53:20:27) vs. saline | 100 mL, twice daily, 6 weeks | No difference in clinical and histological activity scores | [85] |

Ratio = acetate:propionate:butyrate.

**Table 3.** Research showing negative effects of butyrate on barrier function and cytoprotection.

| Model | Result | Mechanism | Ref. |
|---|---|---|---|
| DSS induced colitis mice with antibiotics pretreatment | Not ameliorate colitis | Decreased IL-6 production by butyrate treatment inhibit intestinal tissue repair and cytoprotection | [12] |
| Caco-2 cells | Low concentration; intestinal barrier ↑ Excessive concentration; intestinal barrier ↓ | Excessive butyrate induces epithelial cell apoptosis | [78] |
| Primary intestinal epithelial monolayer cultures from UC patients + TNF-$\alpha$ and IFN-$\gamma$ | Intestinal barrier ↓, IL-8 mRNA ↑ many inflammatory protein ↑ | unknown (There is a possibility that butyrate suppresses the expansion of proliferating cells under inflammatory mediators) | [76] |
| Mouse and human colonic and small intestinal crypts | Suppressing colonic epithelial stem/progenitor cell proliferation, delayed wound repair | By HDAC inhibition, promoter activity for the negative cell-cycle regulator Foxo3 was increased | [77] |

## 8. Conclusions

Therapeutic approaches targeting the gut microbiota may be of great importance for improving treatment outcomes. Butyrate monotherapy is unlikely to cure refractory IBD; however, its plays an adjunctive role in correcting immune abnormalities and epithelial barrier function in the colon, thus enhancing the effects of other drugs. Since the correction of abnormalities in the gut microbiota and appropriate levels of luminal butyrate are necessary for achieving a therapeutic effect, microbiological metabolism and other factors should be considered when conventional therapy has been refractory. The development of a simple marker for easily identifying patients who need gut microbial profile correction is expected. Combining advanced medical therapy with the appropriate restoration of microbial diversity will lead to better outcomes in patients with IBD.

**Author Contributions:** Conceptualization, S.O. and H.S.; writing—original draft preparation, S.O. and H.S.; writing—review and editing, S.O. and H.S.; visualization, S.O.; supervision, H.S.; project administration, H.S. All authors have read and agreed to the published version of the manuscript.

**Funding:** This research received no external funding.

**Institutional Review Board Statement:** Not applicable.

**Informed Consent Statement:** Not applicable.

**Data Availability Statement:** Not applicable.

**Acknowledgments:** We thank all the staffs of our laboratories for their support of this work.

**Conflicts of Interest:** The authors declare no conflict of interest.

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
