# Peer review of "Uptake and Advanced Therapy of Butyrate in Inflammatory Bowel Disease"

_2673-5601, doi:10.3390/immuno2040042_

Round 1
Reviewer 1 Report
The manuscript in my opinion is very interesting,
as the authors wrote ,butyrate utilization and uptake are impaired in IECs and its role in the treatment of IBD coul be useful.
Mauscript is well written.
I suggest to add a table with the main papers so the reviewers can focus immediatly the most important topic.
Reviewer 2 Report
In this article, Ota et al. review the role of butyrate in inflammatory bowel disease (IBD). The article is well-written and sheds light on butyrate absorption and metabolism in IBD patients and their impact on drug therapy.
The authors need to address the following concerns to make the article acceptable:
- Although the article touches upon most of the critical aspects of butyrate metabolism, the primary concern is that all the topics have been just briefly touched upon without any comprehensive discussion. For every topic, the authors need to provide details of in vitro and in vivo studies. For e.g., the role of MCT1 in IECs has been extensively studied and well-established through several models. However, the authors highlight only a few selected studies and base their conclusions. They need to provide more evidence and further discuss the topic. Similarly, they need to be comprehensive for all the other topics.
- To improve the flow of the article, a brief background on IBD pathogenesis, different players involved in it, and then links to butyrate would be useful.
- In the section "How does the gut microbiota influence the response to advanced therapies?" the authors should discuss how butyrate alters gut microbiota composition and affects therapies. Conversely, they must also discuss if treatments alter gut microbiota and butyrate metabolism.
- The authors need to add an additional section (it could be the conclusion section) discussing the implications of the butyrate literature on the diagnosis and/or treatment of different disorders.
- A table or a figure on the controversial role of butyrate would enhance ease of understanding.
- The conclusion section does not clarify the role of butyrate, which is the article's main topic.
Reviewer 3 Report
In the manuscript “Butyrate uptake and advanced therapy in inflammatory bowel disease”, authors have written abstract in well convincing manner. A well written of introduction with relevant background. However following points need to be addressed.
1. In first paragraph of page 2, authors have described role of butyrate and immune system partially. Authors should introduce role of butyrate in context of immuno-inflammation.
2. Title should be redefined as some part is mismatched with introduction. For example, “butyrate uptake and advance therapies of butyrate…….”
3. In section 4, second paragraph is not written in details and confusing with context of cAMP pathways
4. In section 7, authors discussed superficially on advance therapy with butyrate and novel therapy targeting SCFA transporters.
5. In section 7, authors discussed superficially worsening role of butyrate in colitis, needs to discuss in detail.
6. Conclusion is not written in broad manner, should be re-write.
Round 2
Reviewer 2 Report
The authors have a done a good job addressing most of my concerns. The article can be accepted in the present form!
Reviewer 3 Report
In revised manuscript, I have no further comments.